# The Effects of Hyperbaric Oxygenation on Oxidative Stress, Inflammation and Angiogenesis

**DOI:** 10.3390/biom11081210

**Published:** 2021-08-14

**Authors:** Silke D. De Wolde, Rick H. Hulskes, Robert P. Weenink, Markus W. Hollmann, Robert A. Van Hulst

**Affiliations:** 1Department of Anesthesiology, Amsterdam University Medical Centers, Location AMC, 1105 AZ Amsterdam, The Netherlands; r.h.hulskes@amsterdamumc.nl (R.H.H.); r.p.weenink@amsterdamumc.nl (R.P.W.); m.w.hollmann@amsterdamumc.nl (M.W.H.); r.a.vanhulst@amsterdamumc.nl (R.A.V.H.); 2Department of Hyperbaric Medicine, Amsterdam University Medical Centers, Location AMC, 1105 AZ Amsterdam, The Netherlands; 3Department of Surgery, Amsterdam University Medical Centers, Location AMC, 1105 AZ Amsterdam, The Netherlands

**Keywords:** hyperbaric oxygen therapy, hyperbaric oxygenation, oxidative stress, inflammation, angiogenesis, neovascularization

## Abstract

Hyperbaric oxygen therapy (HBOT) is commonly used as treatment in several diseases, such as non-healing chronic wounds, late radiation injuries and carbon monoxide poisoning. Ongoing research into HBOT has shown that preconditioning for surgery is a potential new treatment application, which may reduce complication rates and hospital stay. In this review, the effect of HBOT on oxidative stress, inflammation and angiogenesis is investigated to better understand the potential mechanisms underlying preconditioning for surgery using HBOT. A systematic search was conducted to retrieve studies measuring markers of oxidative stress, inflammation, or angiogenesis in humans. Analysis of the included studies showed that HBOT-induced oxidative stress reduces the concentrations of pro-inflammatory acute phase proteins, interleukins and cytokines and increases growth factors and other pro-angiogenesis cytokines. Several articles only noted this surge after the first HBOT session or for a short duration after each session. The anti-inflammatory status following HBOT may be mediated by hyperoxia interfering with NF-κB and IκBα. Further research into the effect of HBOT on inflammation and angiogenesis is needed to determine the implications of these findings for clinical practice.

## 1. Introduction

Since the adjunctive use of hyperbaric oxygen therapy (HBOT) was first described in 1879 [1], it has been further explored and is nowadays a widely accepted treatment in several diseases, such as delayed radiation injury, diabetic foot ulcers, carbon monoxide poisoning, decompression sickness and arterial gas embolism [2]. The Undersea and Hyperbaric Medical Society (UHMS) describes HBOT as an intervention whereby patients breathe near 100% oxygen while being pressurized to at least 1.4 atmosphere absolute (ATA) in a hyperbaric chamber [1]. Currently, the UHMS has accepted 14 indications for HBOT [3], yet new applications of HBOT have been described, including preconditioning for surgery [4,5,6,7].

Several cohort studies and randomized controlled trials, executed in different surgical procedures (e.g., abdominoplasty and pancreaticoduodenectomy), reported lower postoperative complication rates and a reduced length of stay on the intensive care unit after preoperative HBOT [4,5,6,7]. As the occurrence of postoperative complications is associated with worse short-term and long-term outcomes [8], a decrease in psychosocial well-being [9] and higher healthcare costs [10], HBOT may prevent those adverse effects of surgery.

To realize this perioperative protective effect, HBOT must be able to prevent infection and increase wound healing. It is likely that oxidative stress, which has been confirmed to be the main effect of HBOT [11], plays an activating role in the mechanisms underlying the therapeutic pathway of preconditioning for surgery with HBOT. An increase in reactive oxygen species (ROS) levels is associated with enhanced pathogen clearance [12]. Furthermore, ROS induce the synthesis of several growth factors, such as vascular endothelial growth factor (VEGF), placental growth factor (PGF) and angiopoietin (Ang) 1 and 2 and recruit stem cells from the bone marrow, which are responsible for neovascularization [13]. However, a frequently mentioned argument against the use of HBOT revolves around the induction of oxidative stress as well, since higher levels of ROS and reactive nitrogen species (RNS) may lead to oxidative and nitrosative damage, mitochondrial aging, genotoxicity and maintenance of (chronic) inflammation [14,15,16].

The aim of this review is to gain more insight into the mechanisms of HBOT by assessing its effect on oxidative stress, inflammation and angiogenesis markers in humans. More insight into these effects of HBOT will predict and underpin the outcome of innovative uses of HBOT and balance its benefits against potential damage. No systematic overview of research into these parameters in human beings has yet been published.

## 2. Methods

A search of the literature was performed in MEDLINE and EMBASE on 2 November 2020. Key terms used in the search were ‘hyperbaric oxygen’ and ‘oxidative stress’, ‘inflammation’, or ‘wound healing’. The results were not restricted as no filters were applied. The detailed literature search can be found in Appendix A (see Table A1 and Table A2).

All studies found were screened on title and abstract by one reviewer (S.D.D.W.), who excluded those studies that met any of the following criteria: (1) absence of abstract, (2) congress abstract, errata or guideline, (3) case report (defined as five or less patients), (4) narrative review, (5) animal research, (6) no treatment with HBOT, or (7) one of the following outcome measures: cure, complication rate, or a disease-specific outcome parameter. The same reviewer assessed the full-text of the remaining studies. The following inclusion criteria were applied: (1) measurement of at least one marker of oxidative stress, inflammation, or angiogenesis before and after HBOT, (2) study in humans (or human material) and (3) English full-text available. EndNote X9 was used to keep track of the screening process.

The included studies were divided into an “in vivo” and “in vitro” group. In vivo studies were performed in a clinical setting in which all subjects were at least pressurized once, whereas in vitro studies obtained human material what was subsequently exposed to HBOT. Information on first author, publication year, investigated parameters and patient (in vivo)/sample (in vitro) characteristics and results (solely of the parameters of interest) were extracted. Outcomes of statistical tests with a *p*-value < 0.05 were considered significant. All information was extracted by hand and documented in Microsoft Excel (v16.0).

## 3. Results

### 3.1. Eligible Studies

The search retrieved 9618 records. After removing duplicates and screening of title and abstract, 216 studies were screened full-text. Finally, 137 articles were included in this review (see Figure 1). Most of the included articles were clinical studies (*n* = 98) and performed in patients with diabetes mellitus and/or non-healing chronic wounds (*n* = 27). Furthermore, 27 articles describing the effect of HBOT in healthy volunteers (including divers) were found. Sixteen included studies reported on other biomarkers than described in Table 1, Table 2 and Table 3 (data not shown) [17,18,19,20,21,22,23,24,25,26,27,28,29,30,31,32].

### 3.2. Oxidative Stress

In total, 74 articles reporting on the effect of HBOT on oxidative stress were found. Subjects mainly received one session of HBOT in a hyperbaric chamber pressurized to 2–2.5 ATA (203–253 kPa), yet in seven studies a wet exposure to hyperbaric oxygen (i.e., a dive) to up to 6 ATA (608 kPa) was employed. Nearly 40% (*n* = 21) of the clinical studies were conducted in healthy volunteers (see Table A3). Catalase, glutathione peroxidase (GPx), malondialdehyde (MDA), nitric oxide synthase (NOS), ROS, RNS, superoxide dismutase (SOD) and thiobarbituric acid reactive substances (TBARS) were the most frequent markers of interest (see Table 1).

A clear stimulating effect of HBOT on ROS (see Table 1) was found. Nonetheless, two out of the three studies assessing hydrogen peroxide described lower concentrations after HBOT [33,42] (see Table A3). NOS and RNS concentrations seem to increase after HBOT as well, although this effect was less pronounced, which can be explained by a repeatedly reported decrease in exhaled nitric oxygen [61,69,70]. Timing of sampling may also play a role, as several articles only noted an increase in inducible NOS or nitrite three hours after the end of an HBOT session [34,49,55].

Not only the presence of NOS, RNS and ROS has been investigated, but also their effects on lipids, proteins, carbohydrates and DNA/RNA (see Table 1). Little research has been done regarding protein and carbohydrate modifications following HBOT, but no effect or a stimulating effect on lipid peroxidation, resulting in MDA and other aldehydes (TBARS), has been reported in various studies. DNA-damaging effects of HBOT were not demonstrated employing the most commonly used DNA-lesion-marker 8-hydroxydeoxyguanosine [146].

Concerning the concentrations of anti-oxidative enzymes that protect against the potentially harmful effects of oxidative stress, such as catalase, SOD and GPx, conflicting results were found (see Table 1). In general, no effect or an indication for an increasing effect of HBOT on the enzyme activity of those antioxidants has been demonstrated. HBOT may have a uniform effect on SOD and catalase, as most of the studies reported increased, decreased, or stable SOD and catalase levels and, thus, no differences in effect of HBOT between these two enzymes [76,80,81,83,85,94,97,100]. However, a difference between SOD and/or catalase concentrations in respectively plasma and erythrocytes has been reported [55,62,63]. Benedetti et al. [80] and Dennog et al. [94] describe no effect of HBOT on the free radical trapping anti-oxidants with an exogenous origin, such as vitamin A, vitamin C and vitamin E [149].

### 3.3. Inflammation

Of the 140 studies included, 58 articles describing inflammatory markers were identified. Most of the research included at least three HBOT-sessions, yet study protocols consisting of 20–40 sessions were common, in particular in articles reporting acute-phase proteins (see Table A4). Popular variables of interest were interleukins (IL) (*n* = 31), acute-phase proteins (*n* = 26) and tumor necrosis factor-α (TNF-α) (*n* = 25) (see Table 2).

Concerning acute phase proteins, a decreasing effect of HBOT on (high-sensitivity) C-reactive protein ((hs-)CRP) was found as 75% (*n* = 12) of the studies investigating (hs-) CRP reported lower concentrations post-HBOT. Strikingly, HBOT may have a stimulating impact on granulocyte-colony stimulating factor and an inhibiting effect on insulin-like growth factor-1, both reflecting a pro-inflammatory state [150] (see Table 2).

No impact of HBOT on most interleukin concentrations (IL-2, IL-3, IL-5, IL-7, IL-9, IL-12p70, IL-13, IL-15, IL-17, IL-18 and IL-22) has been demonstrated, although Hao et al. [111] reported a decrease in IL-12p40 levels (see Table A4). Concerning the proinflammatory interleukins, a potentially inhibiting effect of HBOT on IL-1β, IL-6 and IL-8 was found, whereas Dhadmodharan et al. [45] suggested an increase in IL-1α levels. On the other hand, a rise in the anti-inflammatory IL-1Ra was reported, alongside a possible inhibiting effect of HBOT on IL-10 and no effect on IL-4. Both results support an anti-inflammatory state (see Table 2) [151].

In line with the outcomes regarding (hs-)CRP and interleukins, an anti-inflammatory effect of HBOT was also shown by decreasing levels of the pro-inflammatory cytokines interferon-γ (IFN-γ), nuclear factor kappa B (NF-κB) and TNF-α (see Table 2). However, HBOT may have an initial pro-inflammatory effect, as some studies described an increase in TNF-α during or shortly after HBOT [87,127,134].

### 3.4. Angiogenesis

Concerning the angiogenesis research, 34 studies were found in addition to the earlier mentioned studies reporting on interleukins, interferons, insulin-like growth factor 1 (IGF-1), NF- κB and TNF-α. Most of the articles described angiogenesis-inducing cytokines or growth factors and were performed in clinical setting (*n* = 20). However, five out of seven studies on downstream effectors of angiogenesis were conducted in vitro (see Table A5). Epidermal growth factor (EGF), extracellular signal-regulated kinase (ERK), (basic) fibroblast growth factor, tumor growth factor-β (TGF-β), VEGF, IFN-γ, IL-6, IL-8, NF-κB and TNF-α (see Table 2 and Table 3) were the only angiogenesis markers reported in at least five articles.

HBOT most likely has a stimulating effect on various growth factors involved in angiogenesis (i.e., EGF, hematopoietic growth factor, keratinocyte growth factor, PGF and VEGF). This effect may only be present shortly after the intervention, since several studies with repeated HBOT sessions described no differences in pre-HBOT values or only a raise after the first session (and not after following sessions) [62,141,147] (see Table A5). Whereas for some angiogenesis-stimulating cytokines, such as stromal cell-derived factor-1α, a similar increasing effect of HBOT was found, no or an inhibiting effect on TGF was seen. HBOT seems not to affect the cytokine receptors (see Table 3).

HBOT decreased matrix metalloproteinases (MMPs) [52,54,72]. According to Niu et al. [52,72], the effect on MMPs is delayed and only manifests after two or three HBOT sessions. Hypoxia-inducible factor-1α (HIF-1α) and NF-κB were inhibited by HBOT (see Table 2 and Table 3), although Anguinano-Hernandez et al. [125] described an increase in NF-κB in the cytosol.

As HBOT causes an increase in angiogenesis-promoting growth factors and cytokines, one would also expect a stimulating effect on the downstream effectors of blood vessel formation. However, inconsistent outcomes were reported (see Table 3). The phosphatidylinositol-3 kinase (PI3K)/AKT pathway was upregulated and the ERK and p38 mitogen-activated protein kinase (p38 MAPK) pathways were downregulated. Therefore, HBOT effects on downstream effectors of blood vessel formation seem to differ depending on the intracellular effector route.

## 4. Discussion

This review is the first to systematically summarize the effect of HBOT on oxidative stress, inflammation and angiogenesis markers in human beings. HBOT increases the levels of oxygen radicals, which induce oxidative stress. An anti-inflammatory action of HBOT was demonstrated by decreasing concentrations of several pro-inflammatory markers. Furthermore, HBOT seems to stimulate the release of angiogenesis-promoting cytokines, including growth factors.

In the light of previous research, reporting a link between oxidative stress and a pro-inflammatory state [152,153,154], it is remarkable that HBOT leads to a more anti-inflammatory state. However, these findings do correspond with studies into the effects of HBOT using thermal imaging, in which a decrease in wound temperature was found [155,156]. This temperature reduction could indicate a local decline in inflammation. This anti-inflammatory effect is likely mediated by the inhibition of NF-κB, a transcription factor for pro-inflammatory genes [157,158,159]. A direct anti-inflammatory action of HBOT seems less probable, since no differences in the concentrations of anti-inflammatory markers (except IL-1Ra) were noted. Although beyond the scope of this review, Yu et al. [160] have shown in an animal model that HBOT decreases the NF-κB concentrations by higher release of IκBα, which is an inhibitor of NF-κB and degrades under hypoxic circumstances [161]. An increase in IκBα along with a decrease in NF-κB after HBOT was also seen in the only study in the current review reporting on IκBα [52]. Therefore, hyperoxia generated during HBOT may stimulates the preservation of IκBα and thereby inhibits NF-κB release, resulting in less gene transcription of pro-inflammatory cytokines and, thus, an anti-inflammatory state despite oxidative stress.

NF-κB is not only a crucial transcription factor in inflammation, but also plays a role, together with HIF-1α, in the induction of angiogenesis. Growth factors and other angiogenesis-promoting cytokines induce new vessel formation by increased expression of pro-angiogenesis genes, which is mediated by NF-κB or (under hypoxia) HIF-1α [162,163]. Since the current review demonstrates an inhibiting effect of HBOT on both transcription factors and little research, with contradicting outcomes, into the downstream effectors of angiogenesis (i.e., PI3K, Akt, p38 MAPK, ERK) has been done, it is unclear how increased levels of pro-angiogenesis growth factors and cytokines actually induce increased tube formation, as shown by Anguiano-Hernandez et al. [125], Lin et al. [130] and Shyu et al. [40]. Thus, further research into the relation between NF-κB, HBOT and the angiogenesis pathways is needed.

Another striking finding concerning angiogenesis is that several articles reported an increase in growth factors only or particularly after the first HBOT session [40,141,147], while it is common to conduct 20–40 sessions for chronic non-healing wounds or radiation-induced tissue injury (indications strongly relying on the angiogenesis effects of HBOT) [2]. Furthermore, Sureda et al. [62] describe, in the only in vivo study assessing the effect of HBOT on growth factors at several time points during follow-up, an increase in VEGF immediately after each session, yet VEGF levels determined pre-session #5 and #20 were similar to the baseline (pre-session #1) value. Those findings possibly suggest a short pro-angiogenesis effect of HBOT. However, due to a shortage of studies reporting on angiogenesis markers on a daily or weekly basis during a treatment protocol including 20–40 sessions, it remains unclear which markers are involved in this short-term effect of HBOT and whether other factors play a role in this angiogenesis process.

The aim of this review was to gather a comprehensive overview of the effects of HBOT on oxidative stress, inflammation and angiogenesis. We must conclude that existing research does not allow for a complete understanding of the physiology underlying new promising treatment modalities for HBOT, such as preconditioning for surgery. Due to the heterogeneity of included patient populations and the inclusion of studies in healthy volunteers, it is difficult to extrapolate findings to the surgical patient in general. Furthermore, this review did not focus on clinical outcomes related to inflammation, angiogenesis and oxidative stress, making it impossible to determine the implications of the described findings in practice. In conclusion, hyperoxia and oxidative stress induced by HBOT affect inflammation and angiogenesis markers, but whether hyperoxia and oxidative stress induce a clinically relevant decrease in inflammation and increase in angiogenesis remains unclear and needs to be further investigated before innovative interventions can be widely applied.

## Figures and Tables

**Figure 1 biomolecules-11-01210-f001:**
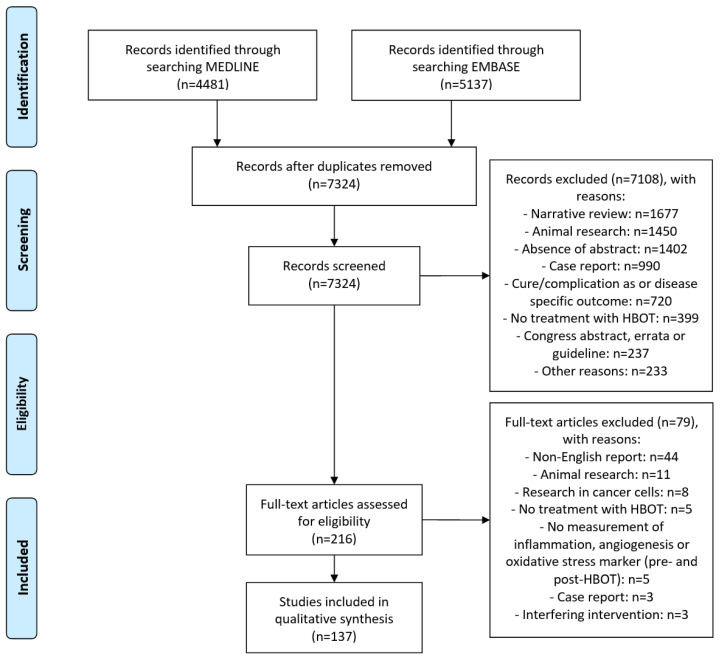
PRISMA Flow Diagram of the selection progress.

**Table 1 biomolecules-11-01210-t001:** The effect of HBOT on oxidative stress markers.

Main Aspect	Associated Markers	Stimulating Effect	No Effect	Inhibiting Effect
Causers of oxidative stress	Reactive oxygen species (including superoxide-ion and hydrogen peroxide)	[33,34,35,36,37,38,39,40,41]	[33,37,38,42,43]	[33,42,43,44]
	Nitric oxide synthase (NOS) (including endothelial NOS and inducible NOS)	[34,45,46,47,48]	[49,50,51]	[52,53,54]
	Reactive nitrogen species (including nitric oxygen, nitrite and nitrate)	[33,34,45,49,55,56,57,58,59,60]	[33,46,61,62,63,64,65,66,67,68]	[53,61,67,68,69,70,71,72,73]
	Hydrobenzoates	[74,75]	[74]	
	Free fatty acid			[53]
	Myeloperoxidase			[34,62]
Lipid peroxidation	Isoprostanes	[76,77]	[78,79]	
	Isofurans		[78]	
	Malondialdehyde	[56,77,80,81,82]	[14,34,49,55,76,83]	[62,84]
	Thiobarbituric acid reactive substances	[63,74,85,86,87]	[33,63,85,88,89,90]	
	Lipid hydroperoxides	[36]	[91]	[92]
	Oxidized low-density lipoprotein		[82]	
Protein peroxidation	Nitrotyrosine	[49]	[64]	
	Advanced oxidation protein products			[77]
Carbohydrate peroxidation	Protein carbonyls	[93]		
	Carbonyl group	[56]		
	Protein carbonyl derivates		[55]	
	Plasma carbonyl proteins		[83]	[82]
DNA/RNA damage	8-hydroxydeoxyguanosine		[83]	[82]
	Tail moment	[38,94,95,96]	[76,97,98]	
	Sister chromatid exchange	[14]		
Gene expression	Nuclear factor erythroid 2- related factor 2	[45]		
Other residues of oxidative stress	Reactive oxygen metabolites	[80]		
	Intracellular calcium concentration			[43]
Antioxidants	Total antioxidant capacity		[91,93,94,99]	
	Catalase	[33,34,43,45,55,62,81,100]	[43,55,62,63,76,83,89,94,97]	[63,80,85]
	Superoxide dismutase	[55,56,81,84,100,101]	[14,33,55,62,63,76,83,89,94,97,102]	[80,85,86]
	Glutathione	[92]	[76,80,83,94]	
	Glutathione disulfide		[76,101]	
	Glutathione reductase		[55,62]	
	Glutathione peroxidase	[34,63,82,85]	[14,55,62,63,76,80,83,89,94,100]	[63]
	Thiols		[88]	[93]
	Vitamin A		[80,94]	
	Vitamin C		[94]	
	Vitamin E		[80,94]	
	Uric acid		[91,102]	[103]
	Heme oxygenase-1	[45,95,96,97]		
	NAD(P)H dehydrogenase [quinone] 1	[45]		

**Table 2 biomolecules-11-01210-t002:** The effect of HBOT on inflammation markers.

Main Aspect	Associated Markers	Stimulating Effect	No Effect	Inhibiting Effect
Acute-phase proteins	(high-sensitivity) C-reactive protein		[88,101,104,105]	[45,53,58,59,60,84,101,103,106,107,108,109]
	Granulocyte-colony stimulating factor	[110]	[110,111,112]	
	Ferritin		[96]	
	Insulin-like growth factor-1	[113]	[67,114,115]	[114,115]
	Albumin	[116]	[102,117]	
Interleukins (IL)	IL-1α	[45]	[111,112]	
	IL-1β		[35,87,111,112,118,119]	[71,72,73,120,121,122]
	IL-1Ra	[111]	[123]	
	IL-1		[4]	[124]
	IL-4		[111,112,118,125,126]	
	IL-6	[62,123,125]	[87,104,105,111,112,118,123,127,128]	[4,35,122,124,129,130,131,132]
	IL-8	[45]	[4,67,105,110,111,112,128]	[46,105,127]
	IL-10	[45]	[106,107,113,114,120,127,128]	[4,105]
Interferons (IFN)	IFN-α		[111,112]	
	IFN-γ	[118]	[111,112,125,126]	[45,133]
Cytokines	Tumor necrosis factor-α	[87,127,134]	[4,45,111,112,128,135,136]	[35,84,105,119,120,121,123,124,129,131,132,133,137,138]
	Nuclear factor kappa B	[139]	[125]	[52,53,125,132,137,140]
Others	Erythrocyte sedimentation rate	[107]		[106,108]

**Table 3 biomolecules-11-01210-t003:** The effect of HBOT on angiogenesis markers.

Main Aspect	Associated Markers	Stimulating Effect	No Effect	Inhibiting Effect
Growth factors/ cytokines	Vascular endothelial growth factor	[45,62,84,125,130,138,141,142]	[50,51,113,120,127,137,143,144,145,146]	[132]
	(basic) Fibroblast growth factor	[45,141]	[102,111,147,148]	[141,143]
	Platelet-derived growth factor		[45,111]	[111]
	Insulin-like growth factor-binding protein	[125]	[67,125]	[67]
	Epidermal growth factor	[45]	[50,111,135,136]	
	Insulin-like growth factor-2		[67]	
	Hematopoietic growth factor	[130,141]	[136]	
	Keratinocyte growth factor	[141]		
	Placental growth factor	[40,141]		[141]
	Tumor growth factor-α		[111]	
	Tumor growth factor-β		[112,147,148]	[115,136]
	Angiopoietin	[84]	[144]	
	Erythropoietin			[145]
	Granulocyte-macrophage colony-stimulating factor		[111,112]	
	Stromal cell-derived factor-1α	[130]		
Cytokine receptors	Tie-2		[144]	
	Erythropoietin-receptor		[37]	
Proteases	Matrix metalloproteinase-3			[52,54,72]
	Matrix metalloproteinase-9			[52]
	Matrix metalloproteinase-13			[52]
Transcription factors	Hypoxia-inducible factor-1α		[125]	[47,130,132]
Downstream effectors	Phosphatidylinositol-3 kinase (PI3K)	[48]		
	AKT	[48]		[53]
	p38 mitogen-activated protein kinase (p38 MAPK)			[44,52,71,72]
	Extracellular signal-regulated kinase (ERK)	[142]	[44]	[52,53,71]

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
