# Peer review of "The Effects of Hyperbaric Oxygenation on Oxidative Stress, Inflammation and Angiogenesis"

_biomolecules, 2021, doi:10.3390/biom11081210_

Round 1

Reviewer 1 Report

The topic of this manuscript falls within the scope of Biomolecules Journal. The topic of the manuscript is very interesting, relevant..

The  data has been provided with vigorous statistical analysis. The Authors have presented sufficient data. The appropriate tables have been provided. The article is easy to read and logically structured.

This review is the first to systematically summarize the effect of HBOT on oxidative stress, inflammation, and angiogenesis markers in human beings. The Authors showed that HBOT-induced oxidative stress reduces the concentrations of pro-inflammatory acute phase proteins, interleukins, and cytokines and increases growth factors and other pro-angiogenesis cytokines.

There are some comments in the reviewer opinion which should be taken under consideration by the Authors:

  1. There are some studies which showed that HBOT has ani-inflammatory action in the treatment of chronic wounds using thermal imaging, please discuss it in the discussion section.

In these studies, the counted isotherm area decreased during therapy. This may suggest a decrease in inflammatory-state areas.

Thermal imaging and planimetry evaluation of the results of chronic wounds treatment with hyperbaric oxygen therapy. Adv Clin Exp Med 2019, 28 (2), 229-236

Evaluation of hyperbaric oxygen therapy effects in hard-to-heal wounds using thermal imaging and planimetry. J Therm Anal Calorim 1411465–1475 (2020).

Thermal Effects of Topical Hyperbaric Oxygen Therapy in Hard-to-Heal Wounds—A Pilot Study. International Journal of Environmental Research and Public Health. 2021; 18(13):6737

The anti-inflammatory effect of HBOT was also presented in the research  “Utilizing Indocyanine Green Wound Imaging In the Management of Hyperbaric Therapy”.

Author Response

Dear reviewers,

Me and my co-authors would like to thank you for the useful feedback on the manuscript ‘The Effects of Hyperbaric Oxygenation on Oxidative Stress, Inflammation, and Angiogenesis’. Your positive comments and indications of points for improvement are much appreciated. Taking your suggestions in consideration, we have adjusted our manuscript. We have uploaded the revised file and the cover letter to the editor can be found in the attachment. Furthermore, a point-by-point response to your comments can be found below.

Yours faithfully,
S.D. de Wolde.

Response to comments Reviewer 1

Point 1: “There are some studies which showed that HBOT has ani-inflammatory action in the treatment of chronic wounds using thermal imaging, please discuss it in the discussion section.

In these studies, the counted isotherm area decreased during therapy. This may suggest a decrease in inflammatory-state areas.

- Thermal imaging and planimetry evaluation of the results of chronic wounds treatment with hyperbaric oxygen therapy. Adv Clin Exp Med 2019, 28 (2), 229-236
- Evaluation of hyperbaric oxygen therapy effects in hard-to-heal wounds using thermal imaging and planimetry. J Therm Anal Calorim 141, 1465–1475 (2020).
- Thermal Effects of Topical Hyperbaric Oxygen Therapy in Hard-to-Heal Wounds—A Pilot Study. International Journal of Environmental Research and Public Health. 2021; 18(13):6737

The anti-inflammatory effect of HBOT was also presented in the research  “Utilizing Indocyanine Green Wound Imaging In the Management of Hyperbaric Therapy”.”

Response 1: We agree that the discussion is more comprehensive if we include the findings of research to HBOT using thermal imaging. Therefore, we have expanded the second paragraph of the discussion (see line 201 – 203). We have referred to two of the mentioned articles (‘Thermal imaging and planimetry evaluation of the results of chronic wounds treatment with hyperbaric oxygen therapy’ and ‘Evaluation of hyperbaric oxygen therapy effects in hard-to-heal wounds using thermal imaging and planimetry’), as we believe that the other articles are less suitable due to use of topical HBOT or focus on the imaging technique and not on clinical outcomes.

Reviewer 2 Report

This review investigates about the the effect of Hyperbaric oxygen therapy
(HBOT) on oxidative stress, inflammation, and angiogenesis to better understand the potential mechanisms underlying preconditioning for surgery. The review is meticulous, precise and deals with all the issues in this area. All the bibliographic entries are appropriate and the tables are easy to read even for a reader distant from these issues but who approaches them for professional reasons. The effort of the authors who conclude the various chapters with their comments but after careful analysis is admirable. Obviously, and the authors themselves point out, further efforts are needed to fully understand this therapy. It is therefore a work to be taken into consideration for a possible publication.

Author Response

Dear reviewers,

Me and my co-authors would like to thank you for the useful feedback on the manuscript ‘The Effects of Hyperbaric Oxygenation on Oxidative Stress, Inflammation, and Angiogenesis’. Your positive comments and indications of points for improvement are much appreciated. Taking your suggestions in consideration, we have adjusted our manuscript. We have uploaded the revised file and the cover letter to the editor can be found in the attachment. Furthermore, a point-by-point response to your comments can be found below.

Yours faithfully,
S.D. de Wolde.

Response to comments Reviewer 1

Point 1: “There are some studies which showed that HBOT has ani-inflammatory action in the treatment of chronic wounds using thermal imaging, please discuss it in the discussion section.

In these studies, the counted isotherm area decreased during therapy. This may suggest a decrease in inflammatory-state areas.

- Thermal imaging and planimetry evaluation of the results of chronic wounds treatment with hyperbaric oxygen therapy. Adv Clin Exp Med 2019, 28 (2), 229-236
- Evaluation of hyperbaric oxygen therapy effects in hard-to-heal wounds using thermal imaging and planimetry. J Therm Anal Calorim 141, 1465–1475 (2020).
- Thermal Effects of Topical Hyperbaric Oxygen Therapy in Hard-to-Heal Wounds—A Pilot Study. International Journal of Environmental Research and Public Health. 2021; 18(13):6737

The anti-inflammatory effect of HBOT was also presented in the research  “Utilizing Indocyanine Green Wound Imaging In the Management of Hyperbaric Therapy”.”

Response 1: We agree that the discussion is more comprehensive if we include the findings of research to HBOT using thermal imaging. Therefore, we have expanded the second paragraph of the discussion (see line 201 – 205). We have referred to two of the mentioned articles (‘Thermal imaging and planimetry evaluation of the results of chronic wounds treatment with hyperbaric oxygen therapy’ and ‘Evaluation of hyperbaric oxygen therapy effects in hard-to-heal wounds using thermal imaging and planimetry’), as we believe that the other articles are less suitable due to use of topical HBOT or focus on the imaging technique and not on clinical outcomes.